# Ultra-High-Capacity Lithium Metal Batteries Based on Multi-Electron Redox Reaction of Organopolysulfides including Conductive Organic Moieties

**DOI:** 10.3390/polym15020335

**Published:** 2023-01-09

**Authors:** Takeshi Shimizu, Naoki Tanifuji, Kosuke Nishio, Yuma Tanaka, Yuta Tsukaguchi, Kentaro Tsubouchi, Fumiya Nakamura, Naoko Shokura, Mariko Noguchi, Hiroki Fujimori, Hiromi Kimura-Suda, Yusuke Date, Kaoru Aoki, Hirofumi Yoshikawa

**Affiliations:** 1Chemistry and Biochemistry Division, Department of Integrated Engineering, National Institute of Technology, Yonago College, 4448 Hikona-cho, Yonago, Tottori 683-8502, Japan; 2Graduate School of Science and Engineering, Chitose Institute of Science and Technology, 758-65 Bibi, Chitose 066-8655, Hokkaido, Japan; 3Department of Chemistry, College of Humanities and Sciences, Nihon University, 3-25-40 Sakurajosui, Setagaya-ku, Tokyo 156-8550, Japan; 4Department of Material Science, School of Engineering Kwansei Gakuin University, Gakuen 2-1, Sanda 669-1337, Japan

**Keywords:** rechargeable battery, organopolysulfide, inverse vulcanization, disulfide

## Abstract

Recently, organic polysulfides have been synthesized as cathode active materials exceeding the battery performance of sulfur. However, the conventional organic polysulfides have exhibited capacities lower than the theoretical capacity of sulfur because the π-organic moieties do not conjugate with the sulfur chains. In this work, the organopolysulfides, synthesized via inverse vulcanization using disulfide compounds, exhibited higher capacities equal to the theoretical capacity of sulfur because of enhanced electronic conductivity based on the conjugation between organic moieties and sulfur chains. Furthermore, the organopolysulfide including 1,3-dhitiol-2-thione moiety exhibited the highest capacity because of the enhanced electronic conductivity. This finding will pave the way to develop next-generation rechargeable batteries.

## 1. Introduction

Lithium-ion batteries (LIBs), which were composed of graphite anodes (capacity of 372 mAh g^−1^) and inorganic cathode active materials, such as LiCoO_2_, LiMnO_2_, and LiFePO_4_ (capacity of 140–170 mAh g^−1^) [1,2], have been used as electric sources in mobile electric devices and electric vehicles because of their capacities and energy densities (250–400 Wh kg^−1^) [3]. Recently, to develop rechargeable batteries with higher capacity and energy density than LIBs for longer-life electric devices and longer-range mobile electric vehicles, anode and cathode active materials have been proposed to develop rechargeable batteries with higher capacity and energy density than LIBs. The candidates with high theoretical capacities have been widely investigated based on Equation (1) (where *n* and *M*_w_ represent reactive electrons and the molecular weight of active materials, respectively) because rechargeable batteries exhibit a lower capacity of either cathode or anode active materials.
(1)$$\mathrm{Theoretical}\;\mathrm{capacity}=\frac{26,800}{M_{w}}n\;$$

One criterion is that redox-active materials were composed of lightweight atoms. The other is that redox-active materials show multi-electron redox reaction. Among the alternative anode materials, lithium metal is one of the powerful materials for the breakthrough of the capacity and energy density limitation of current LIBs because of its light-weight, ultra-high theoretical capacity of 3860 mAh g^−1^ and low redox potential (–3.04 V vs. standard hydrogen electrode) [4,5]. On the other hand, conventional inorganic cathode active materials have not reached a capacity of more than 200 mAh g^−1^ because of heavy-weight metal ions and one-electron redox reaction.

As the candidates for cathode active materials of high-capacity batteries, compounds with disulfide (S-S), such as organodisulfides, organosulfides, and sulfur ring S_8_, have been used. They have been studied as alternative cathode materials because S-S bonds, composed of lighter element S than transition metals, show two-electron redox reaction reversibly. Organodisulfides [6] and organosulfides [7] have been synthesized as cathode active materials with high capacities between 300–1200 mAh g^−1^ since Visco and co-authors reported the electrochemical redox reaction of organodisulfides and S-S bond formation (oxidation)/cleavage (reduction) [8] (Equation (2), where R represents organic moieties).
(2)$${R-S-S-R+2\mathrm{Li}}^{+}{+2e}^{-}\;⇄\;R-S-\mathrm{Li}+\mathrm{Li}-S-R$$

In particular, lithium-sulfur (Li-S) batteries, using sulfur ring S_8_ and lithium metal as the cathode and the anode active materials, respectively, have been investigated as one of the next-generation batteries because of the high theoretical capacity of sulfur ring S_8_ (1675 mAh g^−1^) based on the 16-electron redox reaction (Equation (3)). [9,10]
(3)$$S_{8}{+16\mathrm{Li}}^{+}{+16e}^{-}\;⇄{\;\mathrm{Li}}_{2}S\;$$

Nevertheless, the intermediate lithium polysulfides (Li_2_S_8_, Li_2_S_6_, and Li_2_S_4_) were dissolved into organic electrolyte continuously and migrate between cathode and lithium metal anode during the discharge–charge process, which results in large, irreversible capacities because the lithium polysulfides react with the lithium metal anode (shuttle effect) [9,10]. To overcome the shuttle effect, lithium polysulfides should be converted into insoluble low-order sulfides (Li_2_S_3_ and Li_2_S_2_) before lithium polysulfides are dissolved into the electrolyte.

Recently, inverse vulcanization methods have been reported for the improvement of the cycle performance of sulfur-based cathode active materials. The conventional inverse vulcanization of organosulfides proceeds through the attack of sulfur radicals in the edge of the opened chain toward the electrophilic sites of organic moieties, such as carbon-carbon double bonds (C=C) in 1,3-diisopropenylbenzene (DIB) [11,12]. The organopolysulfide discharge products, such as DIB and styrene units, suppress the irreversible deposition of the insoluble lower-order lithium sulfides (Li_2_S_3_, Li_2_S_2_, and Li_2_S), and as a result, compared to elemental sulfur, the cycle performance is enhanced by the suppression of mechanical damage during discharge–charge process [13,14]. However, the π-organic moieties cannot contribute to enhancement of electronic conductivity of the organopolysulfides without conjugation between π-organic moieties and polysulfide chains. This results in capacities of organosulfides that are lower than the theoretical capacity of elemental sulfur. To overcome the low capacities, it is necessary to develop the synthesis of organopolysulfides in which polysulfide chains are directly bonded to π-organic moieties.

In this work, we report the battery performances of the organopolysulfides synthesized via inverse vulcanization using disulfide compounds from the viewpoint of inverse vulcanization (synthesis) and redox reaction (electronic conductivity). As shown in Figure 1, the disulfide conjugated with π-organic moieties were cleaved easily by the attack of sulfur radicals, and the inverse vulcanization proceeded repeatedly [15,16]. This strategy is effective for increasing the loading of ring sulfur, meaning that the organopolysulfides with high capacities based on the redox reaction of the long sulfur chain were easily obtained. In addition, the electronic-conductive organopolysulfides were obtained by using electronic-conductive 1,3-dithiol-2-thione [17,18]. Moreover, the hybridization between cathode active materials and conductive carbon materials, such as carbon nanotube (CNT) and graphene, has been reported as a method to improve the battery performance [19,20]. In this study, π-organic moieties interact with the π-conjugated ring of conductive carbon materials, suggesting that organopolysulfides would exhibit a more excellent battery performance.

Herein, this work is composed of two sections: (1) synthesis the copolymer-type organopolysulfides, including two organic moieties, namely *m*-phenylene and 1,3-dithiol-2-thione, and (2) evaluation of the sulfur/phenyl/1,3-dithiol-2-thione copolymer-type organopolysulfide as the cathode active materials in comparison to those of the sulfur/phenyl copolymer-type organopolysulfide and sulfur/phenyl/1,3-dithiol-2-thione copolymer-type organodisulfide.

## 2. Materials

All materials were used without further purification. Dimethyl disulfide (>99.0%), sodium sulfinate (>98%), iodine (>99.8%), and 1,3-dithiolbenzene (99%) were purchased from Sigma-Aldrich (Tokyo, Japan). Elemental sulfur (98%) was purchased from WAKO (Osaka, Japan). Bis(tetrabutylammonium) bis(1,3-dithiole-2-thione-4,5-dithiolato)zinc complex (BTBA-BDTD Zn complex, 98.0%), 1,3-benzenedithiol (>95.0%), *p*-toluenesulfonyl chloride (>99.0%), and *p*-toluidine (>99.0%) were purchased from Tokyo Chemical Industry Co., Ltd (Tokyo, Japan). Tetraethylene glycol dimethyl ether (TEGDME, >99%) was purchased from Sigma-Aldrich.

## 3. Methods

### 3.1. Nucler Magnetic Resonance (NMR)

Proton NMR (^1^H NMR) and carbon NMR (^13^C NMR) charts were recorded on BRUKER AVANCE III HD NMR spectrometers (Yokohama, Japan). ^1^H NMR measurements were performed at 400 MHz using tetramethylsilane (TMS) and CDCl_3_ as a standard material and solvent, respectively. Proton chemical shift values were recorded in parts per million (ppm, δ scale) downfield from TMS (δ 0) and CDCl_3_ (δ 7.27). ^13^C NMR charts were recorded at 101 MHz and also were performed using tetramethylsilane (TMS) and CDCl_3_ as a standard material and solvent, respectively. Carbon chemical shift values are also reported in ppm (δ scale) downfield from TMS (δ 0) and CDCl_3_ (δ 77). Data are presented as chemical shift, multiplicity (s = singlet and m = multiplet), and signal area integration in natural numbers.

### 3.2. Synthesis

#### 3.2.1. Syntheses of Asymmetric Organodisulfides

The asymmetric organodisulfides were synthesized via solvent-free reaction of thiol thiosulfonic acid ester **1**, obtained by cross-coupling reaction of dimethyl disulfide (1.88 g, 20.0 mmol) and sodium sulfinate (9.85 g, 60.0 mmol) with iodine (10.2 mg, 40.2 µmol). As shown in Figure 2, the asymmetric organodisulfide 2a was synthesized by grinding 1,3-benzenedithiol and **1** in a mortar. The asymmetric organodisulfides 2b were synthesized from BTBA-BDTD Zn complex (1.41 mg, 1.50 µmol) by using the same method. **1**: ^1^H NMR (Appendix A) δ 2.52 (s, 3H), 7.52–7.62 (m, 2H), 7.62–7.68 (m, 1H), 7.88–7.98 (m, 2H); ^13^C NMR (Appendix A) δ 18.06, 127.07, 129.29, 133.73, 143.68. **2a**: ^1^H NMR (Appendix A) δ 2.46 (s, 6H), 7.26–7.43 (m, 3H), 7.68–7.74 (m, 1H), ^13^C NMR (Appendix A (a) and (b)) δ 22.95, 125.17, 129.49, 138.28. **2b**: ^1^H NMR (Appendix A) δ 2.62 (s, 6H), ^13^C NMR (Appendix A) δ 24.01, 138.47, 210.64.

#### 3.2.2. Syntheses of Organopolysulfides

The organopolysulfides **3a** and **3ab** were synthesized via inverse vulcanization using **2a** and **2b** (Figure 3). **3a** was obtained as follows: Elemental sulfur (1.03 mg, 32.2 µmol) was added to a glass test tube equipped with a magnetic stir bar and heated to 120 °C in an aluminum block until a clear, orange-colored molten phase was formed. **2a** was then directly added into the molten sulfur medium. The resulting mixture was stirred at 120 °C for 8–10 min, which resulted in vitrification of the reaction media. After allowing the reaction mixture to cool to room temperature, the product was taken directly from the test tube using a metal spatula. **3ab** was obtained by the above inverse vulcanization method using **2a** (469 mg) and **2b** (581 mg).

#### 3.2.3. Syntheses of organodisulfide polymers

The organodisulfide polymer **3’ab** was obtained via two-step synthesis (Figure 4). [21] First, *p*-toluene (15.2 g, 80 mmol) was added to BTBA-BDTD Zn complex (4.71 g, 5 mmol) in acetone (91.9 g). After stirring the solution at room temperature for 4 h, the yellow, crystalline precipitation **2’b** (1.79 g, 3.5 mmol) was obtained by filtration and dried at room temperature. Second, the yellow, amorphous solid **3’ab** was obtained by grinding **2’a** (77.7 mg, 0.55 mmol) and **2’b** (304.4 mg, 0.6 mmol) in a mortar for 5 min. **3’ab** (116.1 mg, yield: 63.1%) was purified through extraction and silicagel chromatography using CH_2_Cl_2_, accompanied by evapolation. **3’ab**: ^1^H NMR (Appendix A) δ 2.62 (s, 6H), 7.26–7.31 (m, 3H), 7.68–7.74 (m, 1H); ^13^C NMR (Appendix A) δ 24.01, 138.47, 210.64.

### 3.3. Characterization of 3ab

#### 3.3.1. Thermal Gravimetric Analysis (TGA) and Differential Scanning Calorimetry (DSC)

To investigate polymerization of **3ab** based on inverse vulcanization, TGA and DSC measurements were undertaken. The TGA measurement was carried out with TG8120 (Rigaku) between 27 and 307 °C at a heating rate of 10 °C min^−1^ under a flow of N_2_ gas (20 mL min^−1^). Aluminum sample pans were used for the measurement. The amount of sample used in TGA measurement was 1.41 mg. As the reference, an empty aluminum pan was used. The DSC measurements were performed at a heating rate of 10 °C min^−1^ from –70 to 40 °C after cooling at 10 °C min^−1^ on a PerkinElmer DSC8500 under a flow of N_2_ gas (20 mL min^−1^). Sealed aluminum sample pans and covers were used for the measurement. The amount of sample used in DSC measurement was 2.25 mg. As the reference, an empty aluminum pan was used.

#### 3.3.2. Diffuse Reflectance Ultraviolet–Visible–Near-Infrared (UV–vis–NIR) Spectroscopy

To identify the aromatic moieties in **3ab**, which was insoluble to organic solvents, diffuse reflectance UV–vis–NIR spectroscopy was performed using a UV-3600 UV–vis–NIR spectrophotometer (SHIMADZU) in the range of 300–800 nm wavelength with calcium sulfate as the standard. The samples, namely monomers (**2a** and **2b**) or **3ab**, were ground with calcium sulfate (WAKO) in agate mortars. The molar ratio of samples and calcium sulfate was 1:100. The spectra were obtained as average of 100 sequential measurements.

### 3.4. Electrochemical Measurements

#### 3.4.1. Battery Fabrication

The thin-film cathode was fabricated as follows. First, an organosulfide (**3a**, **3ab**, or **3’ab**) was added into carbon black (Toka black 5500, Tokai carbon), and it was ground in an agate mortar for 30 min. After that, polyvinylidene difluoride (PVDF, Kishida chemicals) binder (Kishida chemicals) was added and ground again for 1 h. The weight ratios of an organosulfide, carbon black, and PVDF were 30:50:20 and 70:20:10. The mixture was stirred in *N*-methylpyrrolidone. The prepared slurry was coated evenly onto aluminum foil (thickness: 20 μm) using a doctor blade technique and dried at room temperature under vacuum overnight at room temperature. The foil was cut into a disc with a diameter φ of 15.95 mm. CR2032 coin-type cells were assembled by using the cathode, lithium foil (φ: 15.50 mm) as an anode, polypropylene film separator, and 1 M LiClO_4_ in TEGDME in an Ar-filled glovebox. Further, to investigate the effect of the π–π interaction between organic moieties in organopolysulfides and CNT [22], the cathodes including CNT were fabricated. The weight ratio of an organopolysulfide (**3a** or **3ab**), carbon black, CNT, and PVDF was 30:40:10:20.

#### 3.4.2. Discharge–Charge Measurement

Galvanostatic discharge/charge tests (voltage: 1.5–3.0 V, current density: 200 mA g^−1^, discharge/charge cycle: 20) were performed at room temperature (approximately 25 °C) on a Hokuto Denko HJ1020mSD8 charge/discharge device. 

## 4. Results and Discussion

**3’ab**, composed of *m*-phenylene disulfide and dithiole-2-thione moieties, was characterized by ^1^H and ^13^C NMR measurements. As shown in Appendix A, the ^1^H NMR chart of **3ab** shows four peaks (δ 1.00–2.50) attributed to the methyl groups in the terminal thiol thiosulfonic acid ester (Appendix A) and three peaks (δ 7.25–7.31) in the aromatic region. Moreover, the peaks in the ^13^C NMR chart of **3ab** were observed at the positions near to those of *m*-phenylene disulfide (**2a**, Appendix A) and dithiole-2-thione (**2b**, Appendix A). These indicate that polymerization of **2’a** and **2’b** occurred through formation of disulfide while maintaining the skeletons of *m*-phenylene disulfide and dithiole-2-thione. Although no ^1^H and ^13^C NMR data of **3ab** and **3a** were obtained due to their insolubility, the skeletons of *m*-phenylene disulfide and dithiole-2-thione in **3ab** and **3a** because of no nucleophilic attack of sulfide toward C=C and C=S bonds. In addition, the peak of S-S bonds at 470 cm^−1^ was observed in the Raman spectrum of **3ab** [23], indicating that the inverse vulcanization proceeded while maintaining the skeletons of *m*-phenylene disulfide and dithiole-2-thione.

Figure 1a,b show the TGA and DSC traces of **3ab**. As shown in Figure 1a, the weight of **3ab** decreased to 40% of the initial state between 120–300 °C, whereas the weight of sulfur dropped near the decomposition temperature of ring sulfur [24]. This means that sulfur chains in **3ab** were decomposed, while the organic group remained. Furthermore, Figure 1b shows a DSC curve of **3ab** measured at a heating rate of 10 °C min^−1^. An endothermic baseline shift due to the glass transition was observed clearly. The glass transition temperature *T*_g_ was ~15.8 °C, which was determined as the temperature at the half height of the heat capacity difference Δ*C_p_* step. This indicates that **3ab** had a polymer network composed of sulfur chain and organic moieties. Furthermore, Figure 1c shows that the peaks were observed in the UV–vis–NIR spectrum of **3ab** at 331 and 460 nm. The one peak was near the peak in the spectrum of **2a** (317 nm). The other was near the peak in the spectrum of **2b** (460 nm). As shown in Appendix A, the diffuse reflectance UV–vis–NIR spectra of **2a** and **2b** were broadened and similar to those of **2a** and **2b** in the CH_2_Cl_2_ solution, indicating the two following points: (1) the aromatic moieties in **3ab** can be detected by the diffuse reflectance UV–vis–NIR spectroscopy; (2) the spectra of **2a** and **2b** were broadened in a constrained environment, indicating that **3ab** included *m*-phenylene and 1,3-dithiole-2-thione moieties. As mentioned above, it was found that **3ab** was composed of sulfur chain, *m*-phenylene, and 1,3-dithiole-2-thione moieties.

Figure 2a–c show the charge–discharge curves of **3a**, **3ab**, and **3’ab** in the lithium batteries during 20 cycles. In the case of **3a** and **3ab**, short (2.5 V) and long (2.0 V) plateaus were observed. These plateaus indicated the initial S-S chain cleavage at 2.5 V accompanied by the subsequent S-S chain cleavage at 2.0 V, and as a result, **3a** and **3ab** exhibited high initial capacities of 750 and 1096 mAh g^−1^, respectively. The initial capacities of **3a** and **3ab** were equal to those of previously reported conductive polymers with sulfur chains (Appendix A) [25,26,27]. The reaction mechanism of **3ab** was revealed by ex situ Raman spectra. As shown in Appendix A, the intensity of the broad peak around 424 cm^−1^ attributed to S-S bonds in **3ab** was small because it was overlapped by that of carbon black. However, the peak around 424 cm^−1^ disappeared and appeared in the discharge and charge state, respectively, indicating that S-S bonds in **3ab** were cleaved and recovered reversibly during the battery reaction. This indicates that **3ab** exhibited a high capacity based on the reversible S-S bond cleavage/formation [23]. Furthermore, the battery including 70 wt% of **3ab** exhibited a capacity of 712 mAh g^−1^, which corresponds to 65.0% of the capacity of 30 wt% of **3ab** (Appendix A). In the case of **3’ab**, the short plateaus were observed at 2.5 and 2.0 V, resulting in the small initial capacity of 309 mAh g^−1^. This explains the two-electron redox reaction per sulfur atom of the sulfur chains in **3a** and **3ab** and the one-electron redox reaction per sulfur atom of the disulfide in **3’ab**. The difference between the capacities of organopolysulfides (**3a** and **3ab**) and those of the disulfide polymer **3a** was caused by the length of S-S chain. However, the capacity retentions of organopolysulfides (**3a**: 59.3% and **3ab**: 62.1%) were lower than **3’ab** (68.9%) after 20 cycles. This indicates that high-order lithium polysulfides (Li_2_S_x_) produced from the sulfur chains in **3a** and **3ab** dissolved into electrolyte during discharge process [6,7] although the disulfide in **3’ab** showed a reversible redox reaction without more dissolution into electrolyte than **3a** and **3ab**. In addition, in comparison to **3a**, **3ab** exhibited the higher capacity because 1,3-dithiol-2-thione could improve the electrical conductivity, which induced a redox reaction of the sulfur chain. Despite the high capacity, the capacity retention of **3ab** was lower than **3a** because more high-order lithium polysulfide produced from **3ab** was dissolved.

The battery performances of **3a** and **3ab** including CNT are shown in Figure 3a,b. Although **3a** and **3ab** including CNT exhibited higher initial capacities of 1082 and 1399 mAh g^−1^ compared to **3a** and **3ab** without CNT, **3a** (661 mAh g^−1^, 61.1%) and **3ab** (622 mAh g^−1^, 44.4%) showed capacity retentions nearly equal to those of **3a** and **3ab** without CNT. This indicates that the π–π interaction between CNT [22] and organic moieties in **3a** and **3ab** improved the electrical conductivities of **3a** and **3ab**, which induced a multi-electron redox reaction of the sulfur chain of **3a** and **3ab**. In addition, the π–π interaction between CNT and dithiole-2-thione moieties, which enhanced the conductivity of **3ab** more effectively than that of **3a**, might induce release of a larger amount of soluble Li_x_S from **3ab**, resulting in the shorter sulfur chain and the lower capacity of **3ab**. However, the capacity retentions of **3a** or **3ab** including CNT were nearly equal to those of **3a** or **3ab** without CNT because organic moieties and CNT could not suppress the dissolution of Lewis basic lithium polysulfides into electrolyte.

## 5. Conclusions

In conclusion, the inverse vulcanization using electronic conductive disulfide compounds could produce the organopolysulfides with a higher capacity than the counterpart organodisulfide polymers. The sulfur/phenyl/1,3-dithiol-2-thione organopolysulfide exhibited a higher initial capacity than that of sulfur/phenyl copolymer-type organopolysulfide. Furthermore, CNT-hybridization could improve the capacity of organopolysulfides based on the interaction between π-organic moieties and the π-conjugated ring of CNT. However, capacity retentions of organopolysulfides were lower than that of organodisulfide polymer. This indicates that high-order lithium polysulfides produced from the sulfur chains during the discharge process dissolved into organic electrolyte, which caused the poor cycle performances [14]. To solve this problem, it is important to introduce Lewis acidic, such as iron phthalocyanine-based complexes [28,29], because it interacts with Lewis basic lithium polysulfides. Work addressing this challenge is currently underway to contribute to the development of next-generation rechargeable batteries with high capacities and excellent cycle performance.

## 6. Patents

The synthetic methods of organopolysulfides and disulfide polymers were subjected to issues of Jpn. Kokai Tokkyo Koho P2015-54834A and Jpn. Kokai Tokkyo Koho JP2012-229329A, respectively.

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
