# Peer review of "Ultra-High-Capacity Lithium Metal Batteries Based on Multi-Electron Redox Reaction of Organopolysulfides including Conductive Organic Moieties"

_polymers, 2023, doi:10.3390/polym15020335_

Round 1

Reviewer 1 Report

The manuscript entitled " Ultrahigh-capacity Lithium Metal Batteries Based on Multi-electron Redox Reaction of Organopolysulfides including Conductive Organic Moieties” reported a organopolysulfides as the cathode materials for lithium batteries. See detailed comments below:

1.     The authors mentioned enhanced electrical conductivity of organopolysulfide due to n-pai conjugation, it would be necessary to actually measure the material level electrical conductivity via two-probe or four-probe methods, some reference samples such as pure S8 would also be important to verify the description.

2.     If the synthesized materials has enhanced electrical conductivity, then it might not be necessary to apply 50wt% carbon conductive agent into the slurry. The authors may try less conductive agent and verify the performance.

3.     3a sample was worse than 3ab sample in Figure 2 but became better in Figure 3. More discussion and explanations are quite necessary here.

4.     The capacity comparison should be within sulfur related cathode rather than LiCoO2. LiCoO2 has advantages in many other aspects rather than specific capacity.

Author Response

Reviewer 1

Comments: The manuscript entitled " Ultrahigh-capacity Lithium Metal Batteries Based on Multi-electron Redox Reaction of Organopolysulfides including Conductive Organic Moieties” reported an organopolysulfides as the cathode materials for lithium batteries. See detailed comments below:

  1. The authors mentioned enhanced electrical conductivity of organopolysulfide due to n-pai conjugation, it would be necessary to actually measure the material level electrical conductivity via two-probe or four-probe methods, some reference samples such as pure S8 would also be important to verify the description.

Answer

Thank you for your comments. We could not measure the electrical conductivity of the organopolysulfide and S8 by four-probe methods. However, as shown in Figure S9, we believe that the organopolysulfide showed the better electrical conductivity because the battery using 70 wt% of organopolysulfide cathode exhibited a sufficient initial capacity of 700 mAh g-1.

  1. If the synthesized materials have enhanced electrical conductivity, then it might not be necessary to apply 50 wt%carbon conductive agent into the slurry. The authors may try less conductive agent and verify the performance.

Answer

Thank you for your comments. We added the discharge-charge curves of the battery including 70 wt% of 3ab (carbon agent: 20 wt%, PVDF: 10 wt%). The plateaus at 2.5 and 2.0 V showed that even a large amount of 3ab showed the redox reaction based on its sulfur chains. Therefore, we modified and added the sentences as follows.

After (Page 6, Line 194-195)

The weight ratios of an organosulfide, carbon black, and PVDF were 30 : 50 : 20 and 70 : 20 : 10.

Before (Page 6, Line 191-192)

The weight ratios of an organosulfide, carbon black, and PVDF were 30 : 50 : 20.

Added sentences (Page 8, Line 259-261)

Furthermore, the battery including 70 wt% of 3ab exhibited a capacity of 712 mAh g-1, which corresponds to 65.0% of the capacity of 30 wt% of 3ab (Figure S11).

  1. 3a sample was worse than 3ab sample in Figure 2 but became better in Figure 3. More discussion and explanations are quite necessary here.

Thank you for your comments. This might be caused by the π-π interaction between CNT and dithiole-2-thione moieties, which enhanced the conductivity of 3ab more effectively than that of 3a. 3ab had the shorter sulfur chains than 3a at the last cycle because a larger amount of soluble LixS was released from the more reactive sulfur chains in 3ab, resulting in the lower capacity. Therefore, we added the sentences as follows.

Added sentences (Page 9, Line 287-290)

In addition, the π-π interaction between CNT and dithiole-2-thione moieties, which enhanced the conductivity of 3ab more effectively than that of 3a, might induce release of a larger amount of soluble LixS from 3ab, resulting in the shorter sulfur chain and the lower capacity of 3ab.

  1. The capacity comparison should be within sulfur related cathode rather than LiCoO2. LiCoO2 has advantages in many other aspects rather than specific capacity.

Thank you for your suggestion. We compared the battery performances of the organopolysulfides to that of other polymers including sulfur chains as shown in Table S1. Also, we modified the sentences as follows.

After (Page 8, Line 251-253)

The initial capacities of 3a and 3ab were equal to those of previous reported conductive polymers with sulfur chains.[25-27]

Before (Page 6, Line 169-170)

The initial capacities of 3a and 3ab, and 3’ab were 5.36, 7.83, and 2.21 times as high as the capacities of LiCoO2 (around 140 mAh g-1).

Added references

  1. Shi Chao, Z.; Lan, Z.; Jinhua, Y. Preparation and Electrochemical Properties of Polysulfide Polypyrrole. Power Sources 2011, 196, 10263–10266.
  2. Dai, S.; Feng, Y.; Wang, P.; Wang, H.; Liang, H.; Wang, R.; Linkov, V.; Ji, S. Highly Conductive Copolymer/Sulfur Composites with Covalently Grafted Polyaniline for Stable and Durable Lithium-Sulfur Batteries. Acta 2019, 321, 134678.

Zeng, S.; Li, L.; Zhao, D.; Liu, J.; Niu, W.; Wang, N.; Chen, S. Polymer-Capped Sulfur Copolymers as Lithium-Sulfur Battery Cathode: Enhanced Performance by Combined Contributions of Physical and Chemical Confinements. J. Phys. Chem. C 2017, 121 (5), 2495–2503.

Reviewer 2 Report

This work reported the synthesis method and battery performances of the organopolysulfides from the viewpoints of inverse vulcanization and redox reaction. It is a new route to improve the performance of Li-sulfur batteries. However, The following issues should be addressed before publishing:

1. The molecular structures of the prepared samples should be further confirmed, such as by using FT-IR and mass spectrometry.

2. The resolution of the figures needs to be further improved.

3. The electrochemical reaction mechanism of the organopolysulfides should be analyzed, or the reaction products after cycling should be detected.

Author Response

Reviewer 2

Comments: This work reported the synthesis method and battery performances of the organopolysulfides from the viewpoints of inverse vulcanization and redox reaction. It is a new route to improve the performance of Li-sulfur batteries. However, The following issues should be addressed before publishing:

  1. The molecular structures of the prepared samples should be further confirmed, such as by using FT-IR and mass spectrometry.

Answer

Thank you for your comments. We added the Raman spectrum as Figure S9. This indicated that the 3ab had both of organic moieties and sulfur chain. Therefore, we modified the sentences and added the sentences as follows.

After (Page 7, Line 217-220)

Although no 1H and 13C NMR data of 3ab and 3a were obtained due to their insolubility, the skeletons of m-phenylene disulfide and dithiole-2-thione in 3ab and 3a because of no nucleophilic attack of sulfide toward C=C and C=S bonds.

Before (Page 7, Line 214-218)

Although no 1H and 13C NMR data of 3ab and 3a were obtained due to their insolubility, inverse vulcanization of 3ab and 3a might proceed while maintaining the skeletons of m-phenylene disulfide and dithiole-2-thione because of no nucleophilic attack of sulfide toward C=C and C=S bonds.

Added sentences (Page 7, Line 220-223)

In addition, the peak of S-S bonds at 470 cm-1 was observed in the Raman spectrum of 3ab [23], indicating that the inverse vulcanization proceeded while maintaining the skeletons of m-phenylene disulfide and dithiole-2-thione.

Added reference

  1. Tao, A.; Zhang, K.; Ma, X.; Song, X.; Liang, J.; Wang, Y.; Liu, Y.; Jin, L.; Tie, Z.; Jin, Z. Building Lithium-Polycarbonsulfide Batteries with High Energy Density and Long Cycling Life. ACS Energy Lett. 2022, 79–89.

  1. The resolution of the figures needs to be further improved.

Answer

Thank you for your comments. We improved the resolution of the figures.

  1. The electrochemical reaction mechanism of the organopolysulfides should be analyzed, or the reaction products after cycling should be detected.

Answer

Thank you for your comments. We added the Raman spectra (Figure S10) of 3ab in the initial, discharged, and charged states. According to the results, we confirmed that the intensity of the broad peak around 424 cm-1 attributed to S-S bonds in 3ab were small because it was overlapped by that of carbon black. However, the peak around 424 cm-1 disappeared and appeared in the discharge and charge state, respectively, indicating that S-S bonds in 3ab were cleaved and recovered reversibly during the battery reaction. Therefore, we added the following sentences in the revised manuscript.

Added sentences (Page 8, Line 253-259)

The reaction mechanism of 3ab was revealed by ex situ Raman spectra. As shown in Figure S10, the intensity of the broad peak around 424 cm-1 attributed to S-S bonds in 3ab were small because it was overlapped by that of carbon black. However, the peak around 424 cm-1 disappeared and appeared in the discharge and charge state, respectively, indicating that S-S bonds in 3ab were cleaved and recovered reversibly during the battery reaction. This indicates that 3ab exhibited the high capacity based on the reversible S-S bond cleavage/formation [23].

Round 2

Reviewer 1 Report

The authors well addressed all the comments. I would recommend that the manuscript be published as it is now.

Reviewer 2 Report

All concerns were addressed.